# Manufacture and Characterization of Cola Lépidota Reinforcements for Composite Applications

**Rémy Legrand Ndoumou** [1,2], **Damien Soulat** [1,*], **Ahmad Rashed Labanieh** [1], **Manuela Ferreira** [1], **Lucien Meva'a** [3] and **Jean Atangana Ateba** [2]

1   Laboratoire de Génie et Matériaux Textiles, Gemtex, Ensait, University of Lille, 59000 Roubaix, France
2   Laboratoire de Mécanique (LME), UFD SI, Université de Douala, Douala P.O. Box 24157, Cameroon
3   Laboratoire d'Ingénierie Civile et Mécanique, Université de Yaoundé I, Yaoundé P.O. Box 8390, Cameroon
*   Correspondence: damien.soulat@ensait.fr

**Abstract:** This study represents the first works on the manufacture of reinforcements for composite applications such as yarns and fabrics using a tropical fiber extracted from the bast of the Cola Lepidota (CL) plant. Different types of products were produced, including twisted and untwisted yarns and woven and quasi-unidirectional fabrics to manufacture composite samples. At each scale, experimental characterizations of textile and mechanical properties were carried out; these properties are compared to those given in the literature concerning natural fiber materials. The results show that the tenacity of twisted and untwisted CL yarns is higher than that of similar products based on flax fibers, which is an important result for the weaveability of these rovings. At the fabric scale, the quasi-unidirectional architecture reduces waviness and shows promising tensile properties compared to woven fabrics. On the scale of composites, these developments made it possible to achieve properties in tensile comparable, particularly in stiffness, to those achieved by composites based on natural fibers. The objectives of this paper are to highlight the advantages and drawbacks of different types of reinforcements, and to present the first characterization of the properties of products based on CL fibers.

**Keywords:** Cola lepidota fiber; woven fabrics; composite reinforcements; mechanical behavior; properties; fabric forming

## 1. Introduction

Natural fibers, especially bast fibers, have drawn attention in recent years because of their potential to replace traditional synthetic fibers as reinforcements in engineering composites with the advantages of being sustainable and environmentally friendly [1,2]. Natural fiber reinforced composites (NFCs) possess many attractive characteristics such as low cost, good specific mechanical properties, excellent thermal and acoustic insulation, less toxic release, and lower energy requirement during manufacturing [3]. The mechanical properties of NFCs are mainly governed by the mechanical properties of the natural fibers and their affinity to polymers. Thus, to obtain the best properties in the loading direction, the ideal reinforcement should maximize the number of fibers aligned in this direction, inside the yarns [4,5], and also at the scale of fabrics. Bensadoun et al. [6] have showed that NFCs based on aligned fabrics have higher mechanical properties compared to randomly oriented fabrics. Consequently, a large number of works have been dedicated to the optimization of the fabric structures of yarns based on natural fibers. Several kinds of reinforcements using with these yarns can be produced, such as woven fabric, with balanced or unbalanced patterns, and braided and knitted fabrics. For woven fabrics, mechanical strength depends on several factors, such as the yarn densities in the warp/weft direction, crimp level, and weaving patterns. Corbin et al. [7] studied the effect of weave pattern and process parameters on the mechanical properties of woven hemp

fabric/epoxy composites made from low-twisted rovings. They showed that plain weave fabric composites achieved the highest tensile strength and modulus, whereas the satin and twill weave fabric composites showed lower values. These woven patterns have also been used before impregnation to optimize, tensile, in-plane shear, and forming properties [8]. Antony et al. [9] studied the effect of fiber content and fabric weave pattern on the mechanical properties of hemp fiber woven fabric/polypropylene composites. They used plain weave and 2/1 twill weave with different areal densities. The 2/1 twill fabric composites performed better in terms of tensile strength, tensile modulus, shear strength, and shear modulus. The influence of crimp level of woven patterns on the mechanical properties of hemp composites has been studied by Karaduman [10]. From hemp yarns of 1000 tex, four different weave types (quasi-unidirectional, plain, basket 2/2, and twill 2/2) were produced to be used as reinforcements of epoxy composites. UD composites show the highest tensile strength, tensile modulus, flexural strength, and flexural modulus in the 0° (yarn) direction. Plain weave fabric composites showed the second highest tensile and flexural strength and moduli, followed by basket 2/2 weave composites. Composites reinforced with twill 2/2 woven fabrics showed the lowest mechanical properties due to high yarn crimp and yarn angle, less balanced structure, and the resulting shear forces during various loading conditions. For woven flax/epoxy composites, Asgarinia et al. [11] reported that fabrics with a lower crimp level can enhance the mechanical strength of the composites. Composites based on UD reinforcements show better stiffness in the fiber direction compared to composites based on randomly oriented fiber reinforcements [12,13]. Shah [14] showed that composites based on unidirectional natural fiber reinforcements offer 2 to 20 times better tensile properties than composites based on nonwoven natural fiber reinforcements. All of these studies concern bast fibers, but the availability of natural fibers varies according to geographical location, which is relevant to the environment because each industrial center can locate available resources, thus meeting the demand for green materials while using new, accessible, and low-cost tropical species [15]. Concerning plant fibers from tropical plants in Central, East, and West African countries, some studies have investigated the identification of the physico–chemical, thermal, and mechanical properties of African star apple leaves [16], bast fibers of Triumfetta cordifolia [17], Rhectophyllum camerunense fibers [18–20], and Cola lepidota (CL) fibers [21,22]. All of these works have been dedicated to the fiber scale and not to the development of yarns and fabrics based on these tropical fibers for composite applications. The manufacturing and characterization of the kind of products produced from Cola lepidota fibers, to the authors' knowledge, has not been previously conducted. For this paper, a weaving loom was used to make two types of reinforcements from twisted and untwisted rovings based on CL fibers: a woven and a quasi-unidirectional fabric with a density of fibers lower in one direction than in the other. These reinforcements were then used to make composite samples. A characterization step was also taken to identify textile and mechanical properties at each scale (yarns/fabrics and composites). This paper is also dedicated to the forming behavior of both these fabrics because few studies have compared deformability between woven (two orthogonal orientations of fibers) and unidirectional or quasi-unidirectional fabrics. Finally, the properties identified experimentally were compared with those of products based on natural fibers, from the literature.

## 2. Materials and Methods

### 2.1. Materials

#### 2.1.1. Rovings and Twisted Yarns

*Cola lepidota* (CL) rovings and yarns were manufactured from CL ribbons extracted from the bast of CL stems, as described in detail in previous studies [21,22]. Due to CL ribbon morphology and structure [22], a process in three steps (ribbons selection, preparation, and twisting) was used to manufacture CL rovings and yarns instead of the conventional spinning process used in the textile industry [23]. The selection step allowed ribbons of lengths of 2 to 6 m, thickness between 0.75 mm and 1.1 mm, and a width range

of 7 to 15 mm to be obtained. They were then joined together with a solution of adhesion with a PVA-based binder, provided by MEDIAN Company (Italy). A twist level of 70 tpm (turns per meter) was finally applied to the roving. Before twisting, the flat rovings were humidified in order to maintain the applied pressure. At this scale, two main products were manufactured:

- *Cola lepidota* rovings, denoted as R1_CL, which are flat, untwisted rovings obtained after the selection step. They have an average width of 10 mm and an approximate thickness of 92.96 $\pm$ 12.74 µm.
- *Cola lepidota* yarns, denoted as Y_CL, obtained from R1_CL previously separated in the mid-plane and twisted on the TWISTEC twister of GEMTEX laboratory.

### 2.1.2. Woven Fabrics

From the previous rovings and yarns, fabrics were woven in the GEMTEX laboratory on a PATRONIC B60 (Handloom Holdings Ltd., Halstead, UK) sampling loom, as shown in Figure 1. Before the weaving step, the preforms were designed using WiseTex® software (Version 2.5) [24]. The design step was followed by the warp preparation (drawing-in, sewing, and tightening). After this stage, two woven fabrics with plain weaving diagrams were produced, as illustrated in Figure 2:

- A plain weave fabric, denoted as Plain_CL, slightly balanced, based on R1_CL with a yarn density of 3 picks/cm in weft and 2.5 ends/cm in warp.
- A quasi-unidirectional fabric, denoted as Quasi-UD_CL, an unbalanced fabric based on Y_CL in the warp direction with a density of 0.5 ends/cm and on R1_CL in the weft direction, with a density of 3.5 picks/cm. In this fabric, the twisted rovings were used to hold the weft rovings in place in order to obtain the maximum quantity.

The choice of plain weave is linked to the fact that although it is the most widely used [6,24]. A quasi-unidirectional structure is highly unbalanced, with a high fiber percentage in one direction relative to the other, therefore the crimp is minimized [12].

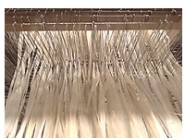

Drawing in

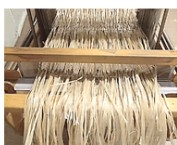

Sewing-in

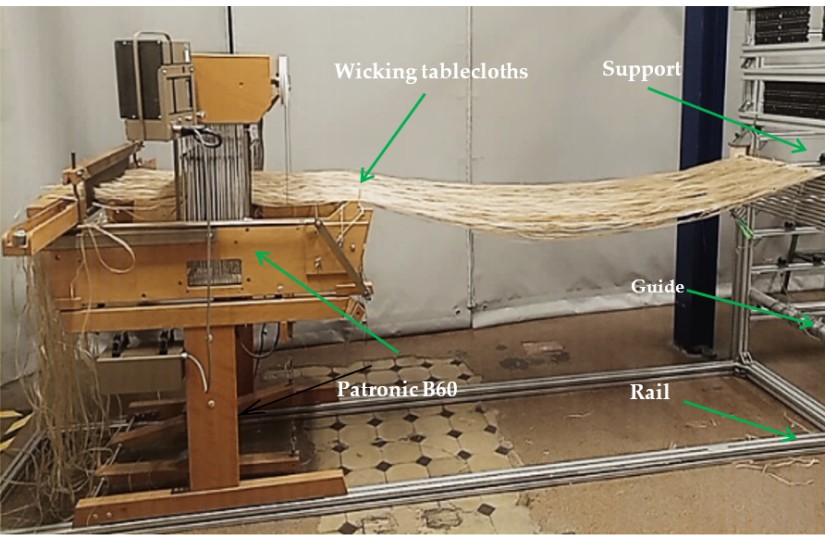

Warp preparation

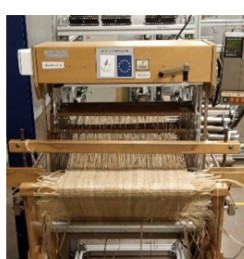

Weaving

**Figure 1.** Weaving process of fabrics.

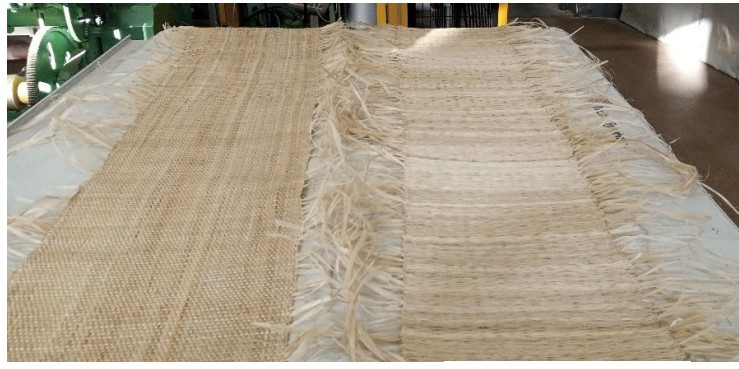

Plain_CL          Quasi-UD_CL

**Figure 2.** Woven and quasi-unidirectional fabrics.

2.1.3. Composite Manufacturing

Plain_CL and Quasi-UD_CL fabrics were used with GreenPoxy 56® epoxy resin and SD Surf Clear hardener from Sicomin® (Chateau neuf les Martigues, France) with a weight ratio of 100/37 to manufacture composite plates using hot pressing. Two biocomposites were produced from two plies of fabrics:

- A quasi-unidirectional composite, denoted as CQUD, manufactured from Quasi-UD_CL fabrics stacked in the weft direction.
- A plain weave ply composite, denoted as CPP, manufactured from Plain_CL fabrics stacked in the warp direction.

Composite plates were manufactured by hot pressing. The fabrics were conditioned at a temperature of 23 °C and a relative humidity of 50% for at least 24 h prior to the composite manufacturing. After this pre-conditioning, the plies were layered-up by hand, impregnated manually with the epoxy resin, and then cured at 60 °C for 6 h under a pressure of 5 bars. The mold was open at the ends, which allowed the excess resin to be removed. Following the manufacturing, the composite plates were conditioned at 23 °C and 50% RH for at least four weeks to reach the moisture content equilibrium and were then cut to dimension of $25 \times 230$ mm$^2$.

*2.2. Characterization Methods*

2.2.1. Rovings and Twisted Yarns Properties

Before identification of the textile and mechanical properties, the bobbins of CL roving and yarn were previously stored at a temperature of $20 \pm 2$ °C and HR% of $65 \pm 2$% for at least 48 h. The linear density was evaluated according to the NF G07-316 standard [25] on twenty samples, 500 mm in length. The twist was measured according to the NF G07-079 standard [26] on ten samples for each type of product. In this paper, the tenacity of CL rovings and yarns was identified using tensile tests conducted before and after the weaving process. For each yarn, 10 samples were tested on an MTS Criterion 45 tensile machine, with a load cell of 1 kN, a gauge length of 200 mm, and a crosshead displacement rate of 100 mm/min, according to the NF EN ISO 2062 standard [27]. After weaving, CL rovings and yarns were manually extracted from each fabric, in both directions for Plain_CL and in the weft direction for Quasi-UD_CL.

2.2.2. Woven Fabrics Properties

- Textile properties

As for the rovings and yarns, before characterization tests, fabrics were stored at $20 \pm 2$ °C and HR% of $65 \pm 2$% for one week. For the identification of textile properties, of thickness, areal density, shrinkage, and air permeability, the NF EN ISO 5084 [28], NF EN 12 127 [29], NF ISO 7211-3 [30], and NF EN ISO 9237 [31] standards were used, respectively. Five samples were tested for each fabric in each direction.

- Mechanical properties

The tensile tests were carried out on fabrics, according to the NF EN ISO 13934-1 standard [32], on an MTS Criterion 45 tensile machine with a load cell of 10 KN and a speed of 100 mm·min$^{-1}$. Five samples of $300 \times 50$ mm$^2$ were tested in each direction (weft and warp) for the Plain_CL samples, and only in the weft direction for the Quasi UD-CL fabrics. The tensile parameters studied were maximal load and strain at maximal load. Tensile loads are presented in N/yarn to avoid the effects of yarn densities.

The bias-extension test (BET) is used to characterize the in-plane shear behavior of biaxial fabrics, as detailed in [8,33–36]. In this paper, this test was only performed on Plain_CL fabric. The BET is a tensile test applied on rectangular woven samples cut at 45° relative to the warp and weft directions, with a criterion size (length/width ratio of sample must be greater than two). In this condition, the distinguishably deformed zones with different in-plane shearing strains can be seen in the specimen, as shown in Figure 3, where the C zone shows total in-plane shearing due to both ends of the yarns being free, the B zones show the half-value of in-plane shearing due to one end of the yarns being fixed, and the A zones are constantly undeformed due to both ends of the yarns being fixed. L and l, the initial length and width of the specimen between the jaws, are 210 mm and 70 mm, respectively.

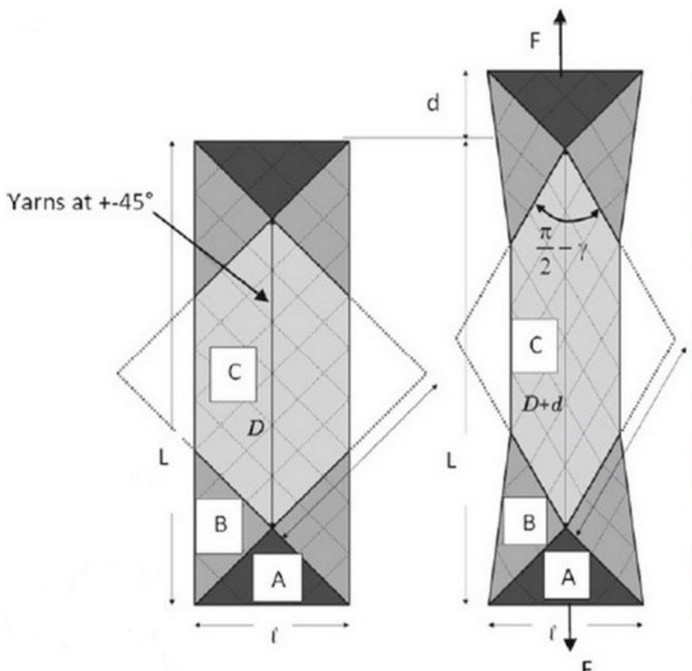

**Figure 3.** Geometry of sample used for the BET and shear zone description.

The bias extension test was conducted on an MTS Criterion 45 tensile machine with a crosshead displacement rate of 20 mm/min and no preload. The evolution of the angle between yarns was obtained via a camera, which was in the stand of the test machine during the BET. To precisely calculate the in-plane shearing angle, $\gamma$, the angles between yarns at five different locations in zone C are averaged. From these pictures, the angle is measured by ImageJ software (Version: 1.53m) after the BET [37]. To avoid potential specimen slippage from the claws in the test machine during the BET, the two ends of the specimens should be reinforced by resin and pressured for 24 h. The measured in-plane shear angle was also compared to those analytically (denoted $\gamma_{th}$) computed with Equation (1), as defined in [33–36], where D represents the length of the pure in-plane shear zone (zone C) and d represents the displacement of the tensile machine.

$$\gamma_{th} = \frac{\pi}{2} - 2\arccos\left(\frac{D+d}{\sqrt{2}D}\right) \tag{1}$$

- Deformability of woven fabrics by forming tests

Forming tests were performed using the specific preforming device developed at the GEMTEX laboratory [8,34,38,39], as seen in Figure 4a. The tested fabric is placed between the upper plate and die because the upper plate is moveable. The blank holder driven by a pneumatic jack applies pressure on the fabric during the preforming. Two punches were used in this study—a hemispherical punch with a diameter of 150 mm (Figure 4b) and a square punch with a dimension of 100 mm (Figure 4c). Only the deformability of the Quasi-UD_CL fabric was tested with the hemispherical punch, while the square punch was used to compare the deformability of both fabrics. A constant pressure of 0.2 MPa on the blank holder and a drawing speed of approximately 45 mm/s were applied. The forming tests were performed using one ply of the woven fabric cut in a square shape with 280 mm sides. The formability of CL fabrics was analyzed after preforming tests with the following parameters: preforming load and fabric draw-in values, shear angles, and defects. Preforming load was obtained using a load sensor located under the punch. The fabric draw-in in each direction (warp and weft) was measured after analysis of the pictures, recorded by the CDD camera placed on the top side of the device, using ImageJ software [36]. The shear angles were computed from the angles between the warp and weft directions and measured in each area on the preform. The defects were identified visually.

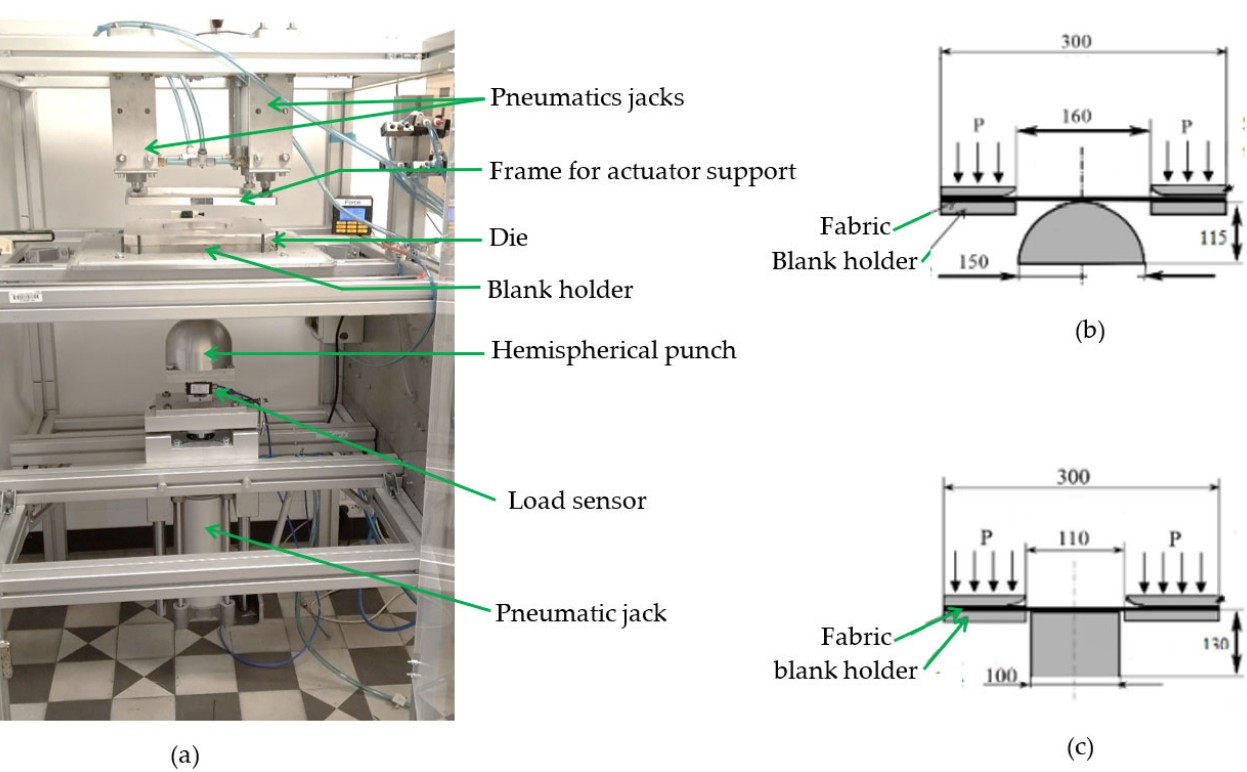

**Figure 4.** Forming test: (**a**) preforming device, (**b**,**c**) unpunched shape dimensions.

### 2.2.3. Composites Properties

The average fiber and porosity volume fractions in each of the composites was determined by the weighing method described in [40]. The CL fiber density was the same as that used in the previous study [22]. Microscopic observations were carried out using a Nikon Eclipse LV-150 optic microscope on $25 \times 100$ mm$^2$ of CPP and CQUD samples.

Tensile tests were conducted on CPP and CQUD samples according to the ASTM D3039-00 standard [41] until failure on an MTS Criterion 45, equipped with a 10 kN cell load. The crosshead speed was fixed at 1 mm·min$^{-1}$. For each composite, six specimens were tested. Digital image correlation (DIC) was used for the recording of the displacement during the test. During a monotonic tensile test in the fiber direction, composites based on natural fiber material were characterized by biphasic behavior [7,12,42]; therefore, two apparent moduli were classically identified. In this study, the tensile modulus was estimated in the strain range of 0.01% to 0.1%.

## 3. Results and Discussion

### 3.1. Roving Properties

Textiles properties of CL rovings and yarns are given in Table 1. Y_CL linear density is around half that of R_CL due to the manufacturing of this yarn, as described in Section 2.1.1. A difference of about 6.66% was observed between the twist level applied to produce the CL yarns and the value measured. This result was attributed to the intrinsic properties of the CL fibers [21,22] and the humidification carried out on the CL ribbons before twisting.

**Table 1.** Textiles properties of manufactured CL rovings and yarns.

| Products | Type | Twist Direction | Linear Density (Tex) | Twist Level (tpm: Turns Per Meter) |
|---|---|---|---|---|
| R1-CL | Flat roving | - | 837.55 ± 145.44 | - |
| Y-CL | Twisted roving | Z | 329.55 ± 61.24 | 70.16 ± 6.58 |

#### 3.1.1. Tensile Properties of CL Yarns and Rovings

The tensile behaviors of CL yarns and rovings are illustrated in Figure 5, with averaged load–strain curves. The evolutions of tangent load–strain are superimposed on the tensile curves. As defined in the literature [5,43,44], tangent load is the slope of the load–strain curve computed for each range of strain. As described in the literature [5,43,44], the tensile curves for CL yarns and rovings have high variability, illustrated by the evolution of the tangent load. The variability of the CL_yarns is lower than that of the rovings, especially at the end of the tests. Twist level increases the cohesion between the fiber bundles inside yarns and decreases the variability, especially near failure.

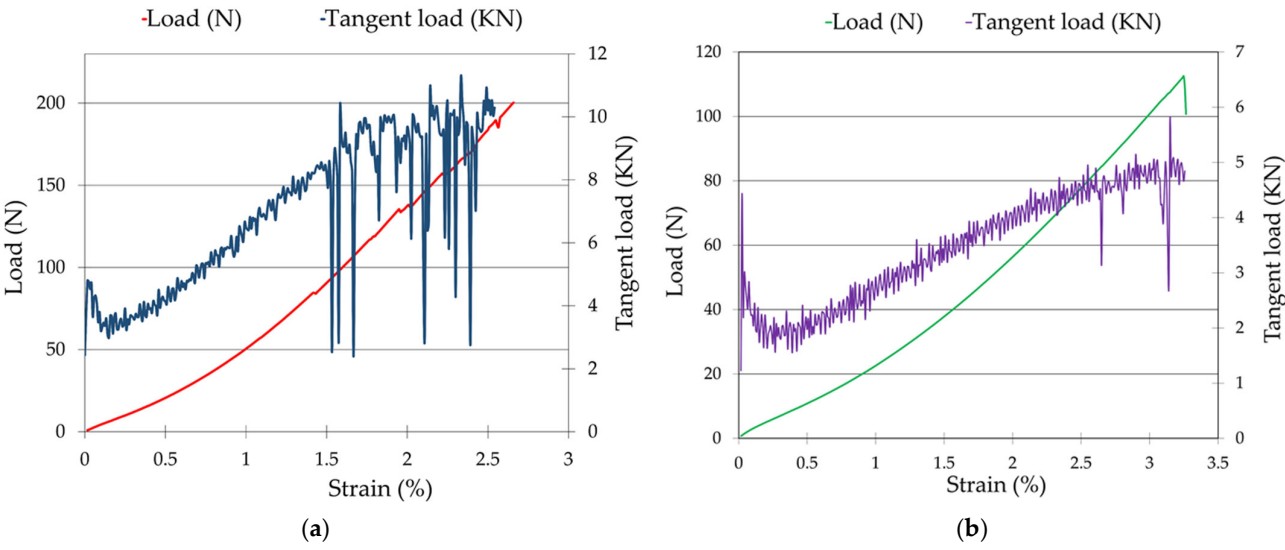

**Figure 5.** Average load/tangent load–strain curves of CL rovings/yarns: (**a**) R1_CL, (**b**) Y_CL.

These tensile curves are characterized by biphasic behavior, with a first phase, at strains below 0.5%, which is characterized by the slight softening. In the second phase, the material behavior is quasi-linear until failure. In a previous study [22], similar behavior was observed in the tensile characterization of CL fibers. Figure 6 shows the comparison of the tensile properties of the tested CL rovings and yarns. To avoid the influence of the linear density, strength properties were compared in tenacity. The twist level applied on CL-Yarns increases tenacity compared to untwisted roving [45]. The values of tenacity reached are sufficient for the weaving process [5,44,46–48]. These results also show that increasing of the twist level increases the strain at break, as shown by Lansiaux et al. on flax roving [48]. The CL-based yarns and rovings developed in this study, without any treatment input, have interesting characteristics compared to the properties of yarns and rovings based on natural fibers available in the literature (Table 2).

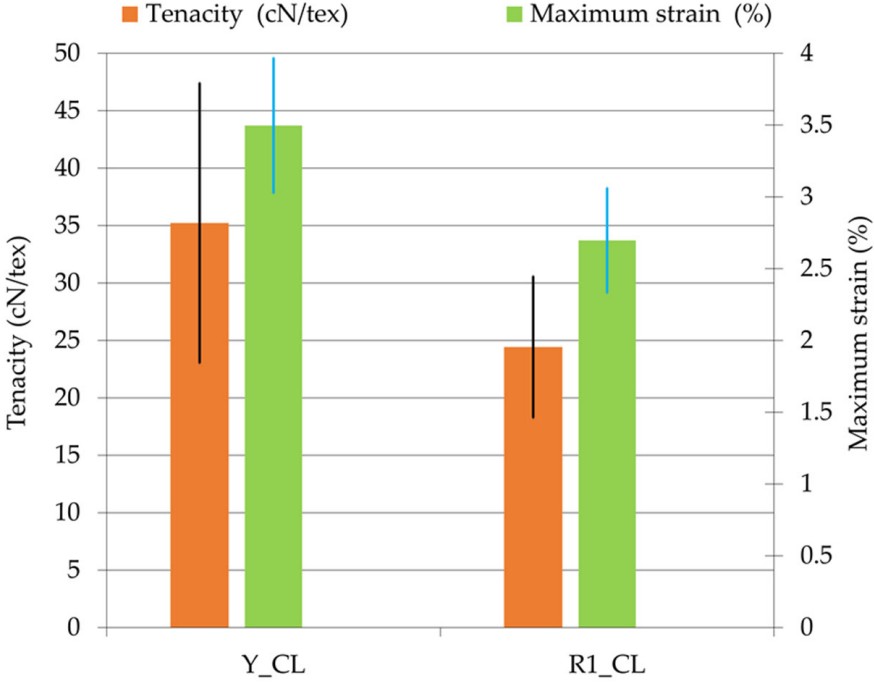

**Figure 6.** Tenacity and maximum strains of CL yarns and rovings.

**Table 2.** Properties of CL rovings and yarns and other natural fiber-based yarns/rovings.

| Name | Raw Material | Twist Level (tpm) | Linear Density (Tex) | Tenacity (cN/Tex) | Strains (%) | Réf |
|---|---|---|---|---|---|---|
| Sisal of Morocco | Sisal | 80 | $3300 \pm 700$ | $20.24 \pm 3.25$ | 6–7 | [49] (yarns) |
| Lincore R500 2016 | Flax | 80 | 500 | 24 | 6.5 | [48] |
| Lincore R1000 2016 | Flax | 70 | 1000 | 21.5 | 11 | (rovings) |
| Roving de Safilin | Flax | 41 | 280 | 4 | - | [43] (rovings) |
| Canapa ST | Hemp | $32 \pm 4.4$ | $334 \pm 26$ | 7–8 | 3.5–4 | [44] |
| Lino ST | Flax | $31.6 \pm 2$ | $370 \pm 49$ | 3–4 | 1.7–2 | (rovings) |
| R1_CL | CL | - | $837.55 \pm 145.44$ | $24.4 \pm 6.17$ | $2.69 \pm 0.36$ | This study |
| Y_CL | CL | $70.16 \pm 6.58$ | $329.55 \pm 61.24$ | $35.22 \pm 12.2$ | $3.49 \pm 0.47$ | |

### 3.1.2. Influence of the Weaving Process on the Tensile Properties of CL Rovings

Some studies in the literature [50,51] have shown that the weaving process may reduce the properties of the yarns and rovings in the fabrics and consequently the properties at

the scales of fabrics or composites. The mechanical properties of CL rovings and yarns extracted from Plain_CL and Quasi-UD_CL fabrics (denoted after weaving) were identified and are compared to those before weaving. These results are presented in Figure 7. For the Plain_CL fabric, the roving R1_CL was kept before and after weaving in the warp and weft directions; for the Quasi-UD_CL fabric, rovings were kept only in the weft direction, after weaving. In the warp direction, the loss in tenacity of R1_CL in the Plain_CL fabric was significant (around 60%) after weaving. In the weft direction, the tenacity of the rovings before and after weaving was similar in both Plain_CL and Quasi-UD_CL fabrics. The consistency of the tenacity of the rovings before and after weaving in the weft direction is attributed to the fact that, in this direction, the rovings were not subjected to the same preparatory draw-in and sewing steps (described in Figure 2), which could damage yarns as in the warp direction with the steps. In the Plain_CL fabric, the strain at break of the roving after weaving increased in both directions (Figure 7b). The increase was more significant in the warp direction, at around 15% compared to the value of 4% in the weft direction. This increase is due to the higher crimp (Figure 7b) compared to the shrinkage, in this fabric. The woven parameters used for the Quasi-UD_CL fabric show that the tenacity and the strain at break of R1_CL remained similar to the values before weaving. Corbin et al. [12] explained this result through the low friction between the rovings in both directions during weaving. Additionally, the shrinkage was very low compared to the value obtained with the Plain_CL (Figure 8b); consequently, the crimp is minimized in this type of fabrics.

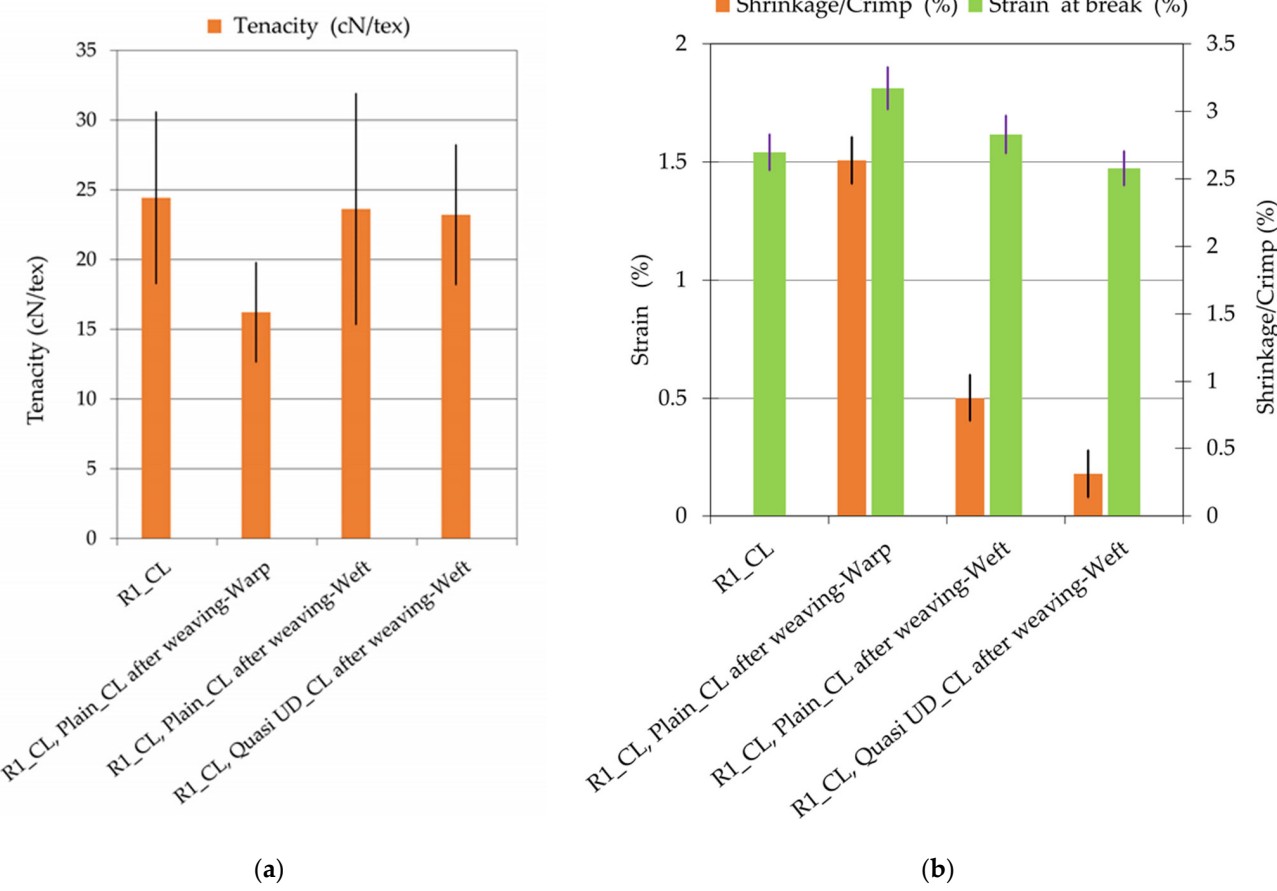

(**a**)                                                                 (**b**)

**Figure 7.** CL roving properties before and after weaving: (**a**) tenacity at break, (**b**) strain at break/crimp and shrinkage.

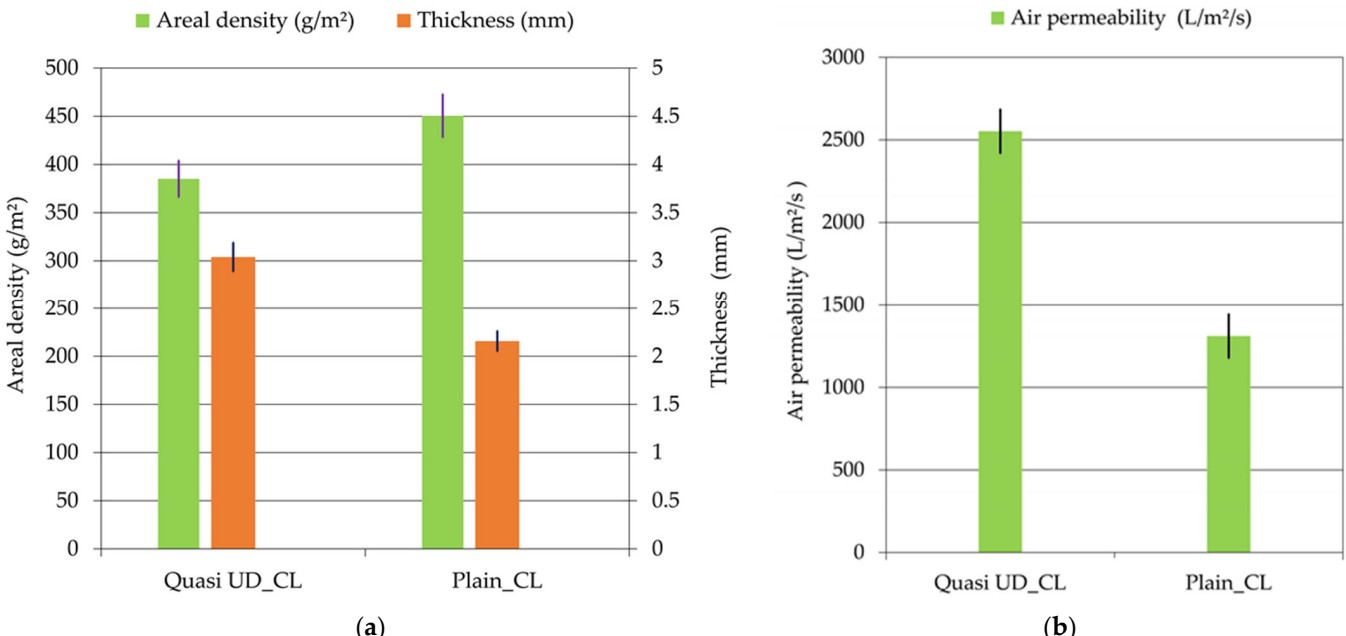

**Figure 8.** Textile properties of fabrics: (**a**) areal density and thickness, (**b**) air permeability.

### 3.2. Fabrics Properties

### 3.2.1. Textile Properties

Figure 8 shows the textile properties measured for the fabrics. Areal densities and thicknesses are given in Figure 8a, and air permeability is given in Figure 8b.

An average value of $450 \pm 25$ g/m$^2$ was identified for Plain_CL fabric, which is 14% higher than the areal density of the Quasi-UD_CL fabric. This difference is due to the linear density of the R1_CL roving used in the warp and weft directions in the Plain_CL fabric and the higher yarn density in the warp direction. The average thickness of the Quasi-UD_CL fabric is $3.04 \pm 0.31$ mm, which is 29% higher than the thickness of the Plain_CL fabric, also due to the higher density of the R1_CL roving used in the weft direction. The difference in air permeability between these fabrics is about 48%. The high level of porosity of the Quasi-UD_CL compared to the Plain_CL is due to its very low warp density and greater thickness.

### 3.2.2. Mechanical Properties

- Tensile properties

The average load–strain curves (with standard deviation illustrated with vertical lines) from the uniaxial tensile tests are shown in Figure 9. Tensile behavior is given for each direction, for the Plain_CL fabric in Figure 9a, and for the weft direction only for the Quasi-UD_CL fabric in Figure 9b. Loads are given in N/yarn to avoid the influence of yarn densities. As described in the literature [5,8,10,52], the tensile response of fabrics is characterized by two main phases before the break. The first phase is non-linear and associated with a gradual reduction of crimp. The second phase is quasi-linear because the yarns are aligned along the load direction. For the Plain_CL fabric, Figure 9a, the linear part begins at strain values of 0.85% and 0.6%, respectively, in the warp and weft directions, which are linked to the crimp and shrinkage values (Figure 7b). Comparatively, for the Quasi-UD_CL fabric with crimp minimized, the linear part begins at a lower strain value of 0.3% (Figure 8b).

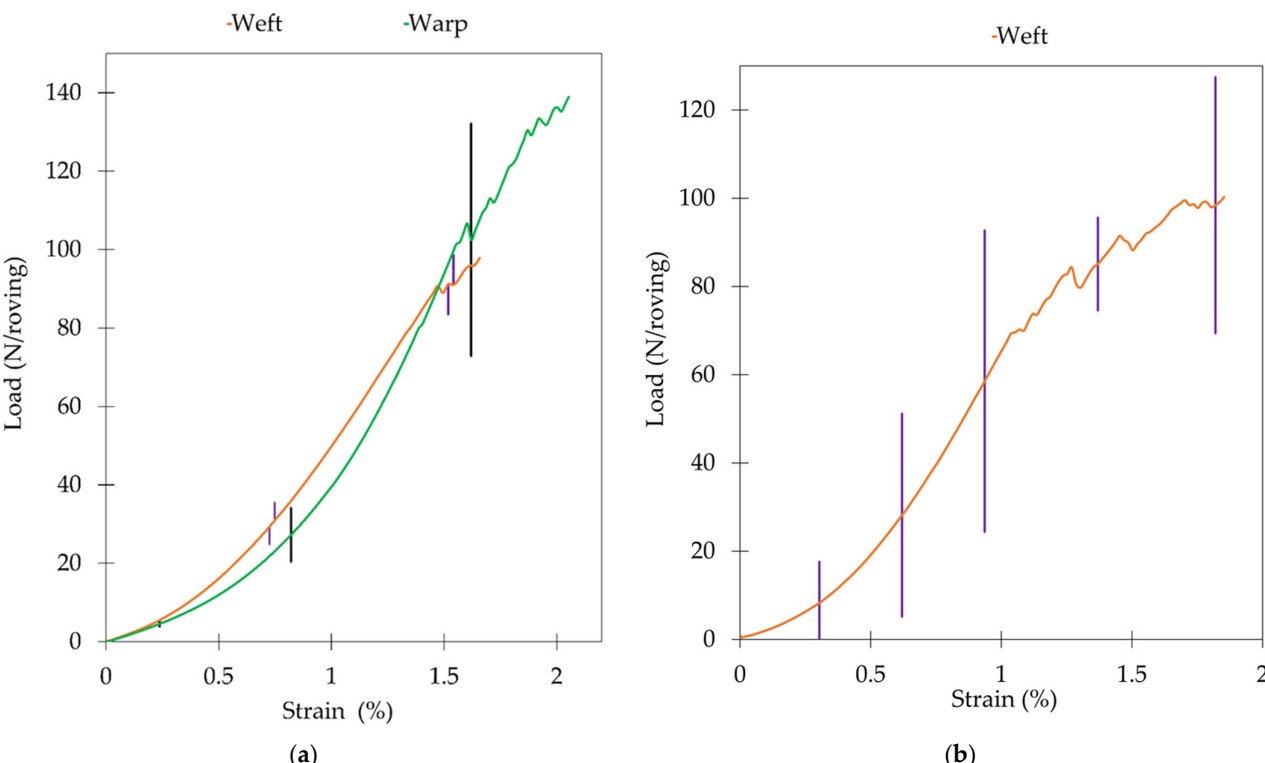

**Figure 9.** Tensile curves of fabrics: (**a**) Plain_CL fabric, (**b**) Quasi-UD_CL fabric.

Concerning the load at break, given in N/yarn, the values reached by the R1_CL rovings inside the fabrics were lower than those identified at the scale of roving (Figure 5a), with a maximum load of 140 N/yarn and 100 N/yarn in the warp and weft, respectively, for the Plain_CL fabric, and around 100 N/yarn for the Quasi-UD_CL fabric. This difference is due to the weave effect. In Figure 10, the values associated with the strain at maximum load of both fabrics, and in the warp/weft directions for Plain_CL fabric, are compared. Due to the stiffness of the fabrics, these values are lower than those at the scale of R1_CL roving. This tendency follows the values identified after weaving, as shown in Figure 8a.

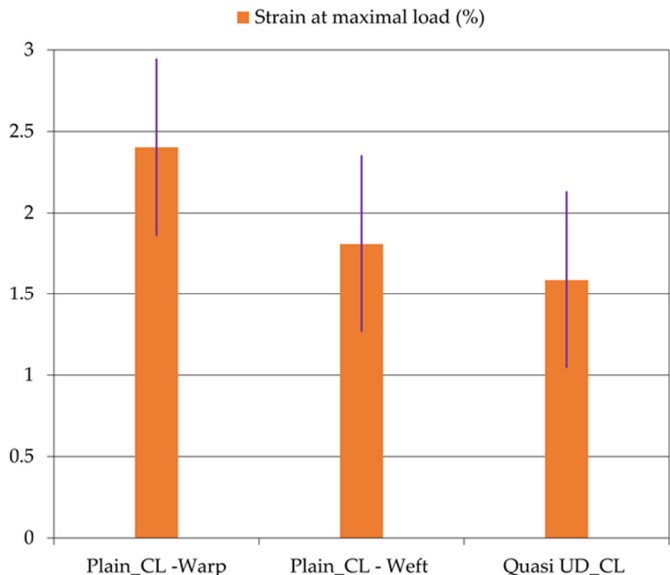

**Figure 10.** Strain at maximum load of fabrics.

- In-plane shear behavior of Plain_CL woven fabric

The resulting load-displacement curve of the BET conducted on the Plain_CL is given in Figure 11a. The Plain_CL fabric has a similar in-plane shear behavior to that of woven flax fabric (with around the same areal density), as characterized by Tephany et al. [53], with an increasing displacement for very small loads due to the rotation of R1_CL rovings. In the second part, the in-plane shear stiffness increased when these tows were in contact. Measured shear angle values are compared with the theorical value computed by Equation (1) in Figure 11b. According to the literature, the in-plane shear locking angle is considered to be reached when the two curves split; beyond this value, the specimen is no longer considered to be in pure shear (slippage or wrinkles) [33–36,54]. For the Plain_CL fabric, this value is around 34°. Corbin et al. [8] performed UBE tests on two plain weave fabrics from flax (FPW) and hemp (HPW) fibers with higher warp/weft densities (6 and 5.8/5 for HPW) than those used to weave the R1_CL rovings. The blocking angles identified were much higher (59° for FPW and 52.7 for HPW) compared to the Plain_CL fabric. On the other hand, the linear densities of the rovings used were much lower (around 300 Tex for flax and 260 for hemp) than that of the CL rovings (837 Tex), which would suggest, despite the lower warp/weft densities used, that there is less space for the roving rotation phenomenon, leading to a smaller blocking angle.

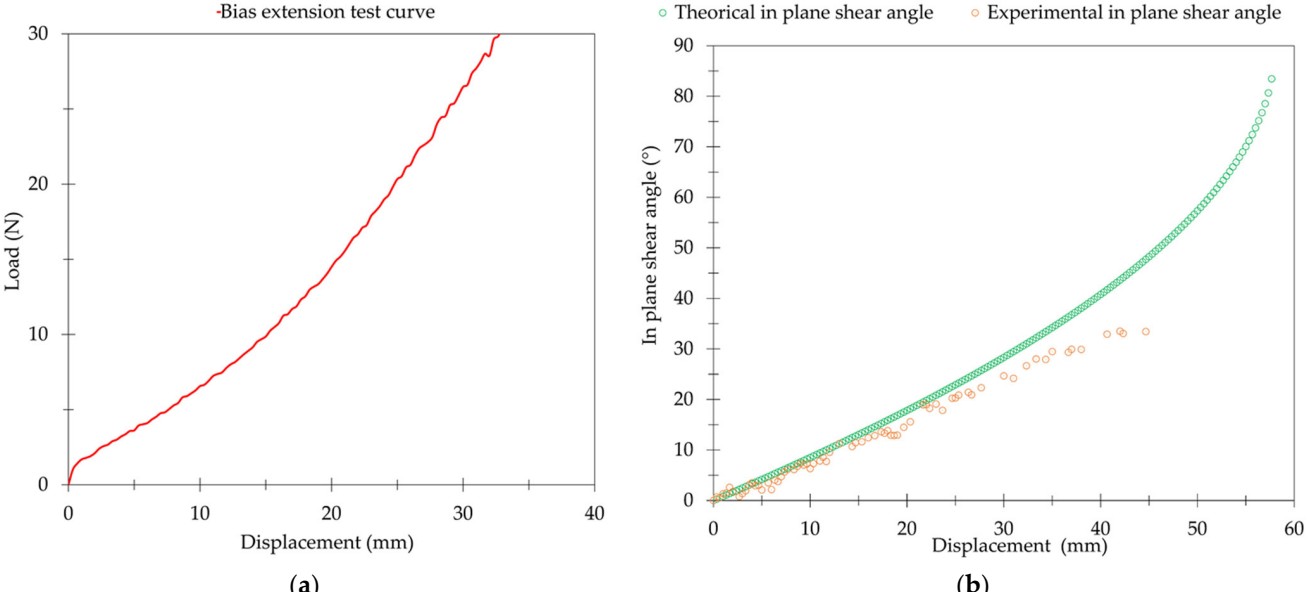

(**a**)  (**b**)

**Figure 11.** Bias-extension test results: (**a**) load-displacement curve, (**b**) theoretical and measured in-plane shear angles vs displacement.

- Forming test

Table 3 gives the maximum preforming load recorded by the load sensor as a function of punch type for a constant pressure of 0.2 MPa applied to the blank holder for both fabrics. Preforming load increases with the friction of the punch and the blank holder. For the square punch, it was 2.82 times higher on the Plain_CL preform than on the Quasi-UD_CL. As discussed in the literature [24,55], for the same orientation of the fabric and with the same punch, this difference is attributed to the higher areal density of the Plain_CL fabric. Due to the complex shape of the square punch, the preforming load was 1.94 times higher than that measured with the hemispherical punch for the Quasi-UD_CL fabric.

**Table 3.** Maximum preforming force on woven fabric.

| Preform | Pressure on Blank Holder (MPa) | $F_{max}$ with Hemispherical Punch (N) | $F_{max}$ with Square Punch (N) |
|---|---|---|---|
| Plain_CL | 0.2 | - | 955 |
| Quasi-UD_CL | 0.2 | 174 | 338 |

The fabric draw-in on the square punch for both fabrics represents the quantity of fabric needed to closely fit the punch shape, as illustrated in Figure 12. The draw-in is measured along the fabric on one side in the warp direction and on another side in the weft direction, and it increases symmetrically from the corner to the middle of the side of the fabric. For the Quasi-UD_CL fabric, draw-in is more significant in the weft direction than in the warp direction, for which the density of twisted rovings (Y_CL) is very low. Table 4 summarizes the values obtained after preforming measured with images of the forming test using ImageJ software. For the Plain_CL fabric, which is balanced, the draw-in is similar in the warp/weft directions. Draw-in with the hemispherical punch on the Quasi-UD_CL fabric was too small to be measured, which explains the values close to zero reported in Table 4. It can be assumed that preforming loads and the shape of this punch were too weak to create tensile strain in these fabrics.

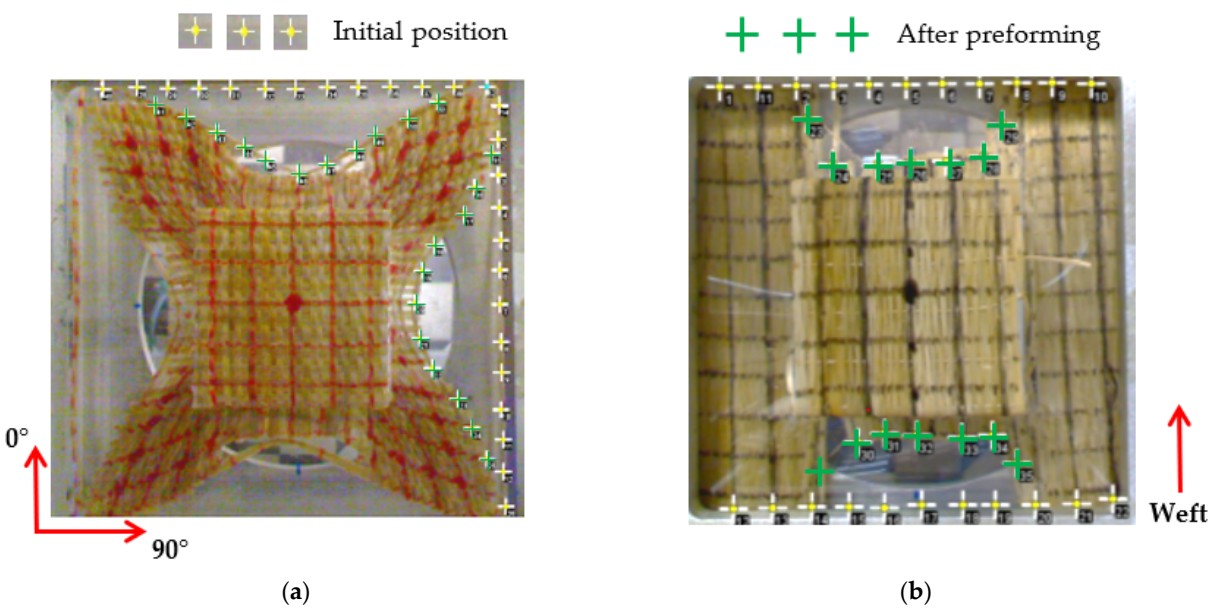

(**a**)    (**b**)

**Figure 12.** Shape and draw-in after preforming by square punch: (**a**) Plain_CL, (**b**) Quasi-UD_CL.

**Table 4.** Maximum draw-in values after forming.

| Preform | Punch | Max Draw-In Weft Direction (mm) | Max Draw-In Warp Direction (mm) |
|---|---|---|---|
| Plain_CL | Square | 51 | 52 |
| Quasi-UD_CL | Square | 41 | ~0 |
| Quasi-UD_CL | Hemispherical | ~0 | ~0 |

Figure 13 shows the values of the shear angles (drawn by colors) in the fabrics at the end of the forming test. With the hemispherical shape, the maximum sheared zones were at the base of the hemisphere (in red in Figure 13b). For the square punch, the maximum values were also reached at the base of the cube, but in the continuity of the four vertical edges (in red in Figure 13a). The zones of maximum value of in-plane shear angles were also discussed by Huang et al. [34] for a woven carbon fabric with a cubic punch, and by Corbin et al. [8] for flax/hemp woven fabrics with a hemispherical punch.

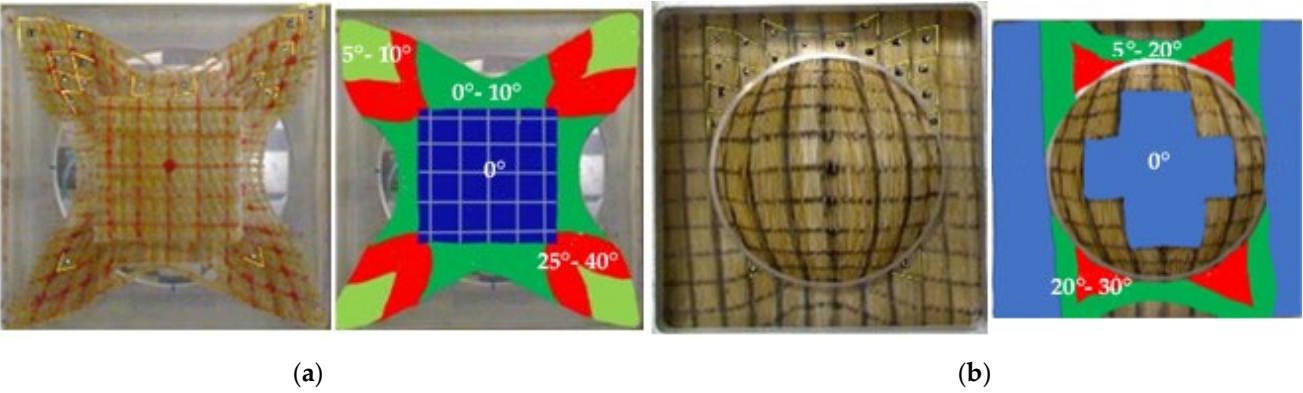

(**a**)                                           (**b**)

**Figure 13.** Shear angle during forming test. (**a**) Plain_CL fabric on square punch (**b**) Quasi-UD_CL fabric on hemispherical punch.

The analysis of the Plain_CL fabric after the forming test shows that the hemispherical punch did not lead to any defects on this fabric. This result was also obtained by Corbin et al. [8] and Labanieh et al. [39] on plain woven fabrics. As mentioned, few papers have reported preforming tests performed on quasi-unidirectional fabrics, particularly for type of defect. Figure 14 shows the sliding of woven tows, especially in the warp direction, where the density of Y_CL yarns is too low to maintain the Quasi-UD_CL structure with the square punch as well, to a lesser degree, in the double curvature of the hemispherical punch.

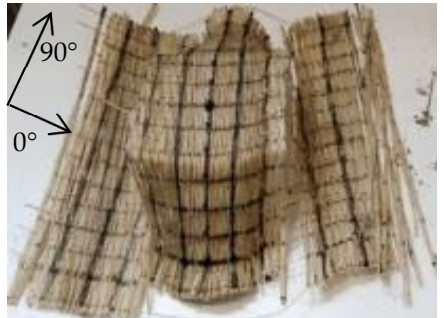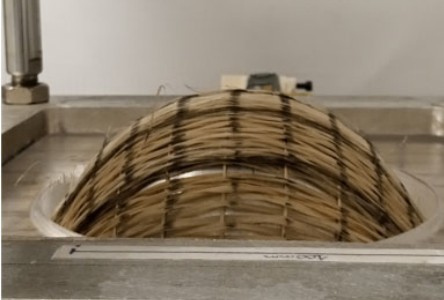

**Figure 14.** Defects on Quasi-UD_CL fabric after forming.

*3.3. Composite Properties*

3.3.1. Fibers and Porosity Contents of Composites

The influence of the manufacturing parameters of composites based on CL fibers, in particularly the pressure, on the quality of impregnation and on the ratio of fibers and porosities is illustrated in Figure 15 and Table 5.

Microstructural views of the cross sections of the composites (CPP and CQUD) manufactured with a pressure of 5 bar show, within the samples and around yarns, few porosities or dry areas without resin (Figure 15). On the other hand, these images do not allow perfect impregnation within the yarns to be confirmed. On the external faces of samples, non-homogeneous areas could explain the high porosity level obtained for this pressure value (Table 5). With an increase of pressure to 25 bar, a significant reduction of void content was observed in contrast to fiber content. It has been shown that, for natural fiber composites, increasing the pressure up to a certain threshold decreases porosity content [56]. Changes in the process parameters also affect the mechanical properties of the final composite material [57].

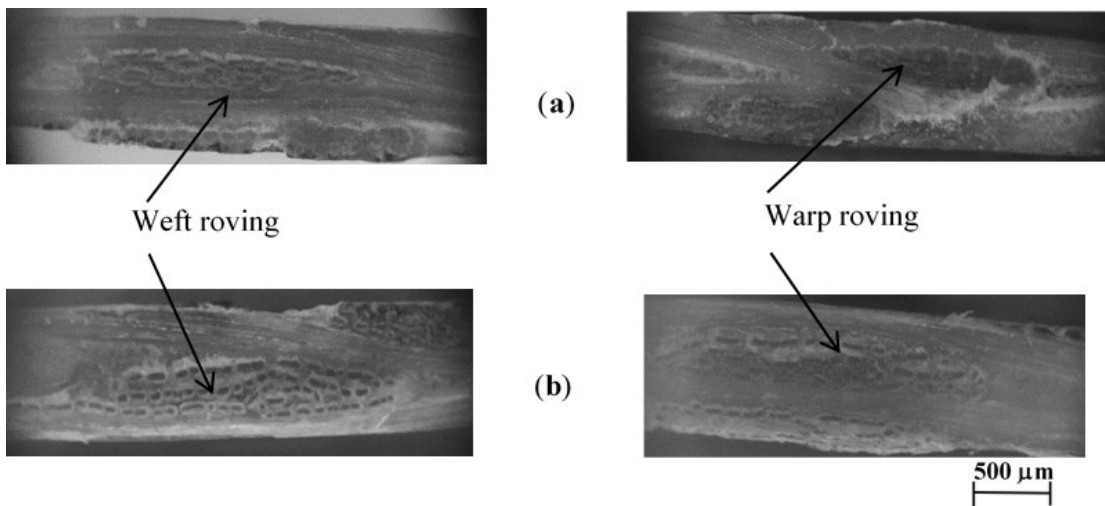

**Figure 15.** Cross-section views of composite (5 bar pressure): (**a**) CPP, (**b**) CQUD.

**Table 5.** Fiber and porosity volume fraction in the composites.

| Composite | Fabrics | Pressure | | | |
|---|---|---|---|---|---|
| | | 5 Bar | | 25 Bar | |
| | | $V_f$ (%) | $V_p$ (%) | $V_f$ (%) | $V_p$ (%) |
| CPP | Plain_CL | 30 | 15.7 | 30.33 | 5.11 |
| CQUD | Quasi-UD_CL | 30.4 | 13.5 | 33.64 | 6.53 |

### 3.3.2. Tensile Properties of Composites

The tensile responses of the four samples (in colors) for each of the two composites—CPP and CQUD—are shown in Figure 16. For the CPP composite, the tensile load was applied in the warp direction of the Plain_CL fabric, while for the CQUD composite, the tensile load was applied in the weft direction of the Quasi UD-CL fabric. The stress–strain curves show a similar trend in both of the composites, with clearly biphasic behavior, as described in the literature [9,12,40,42].

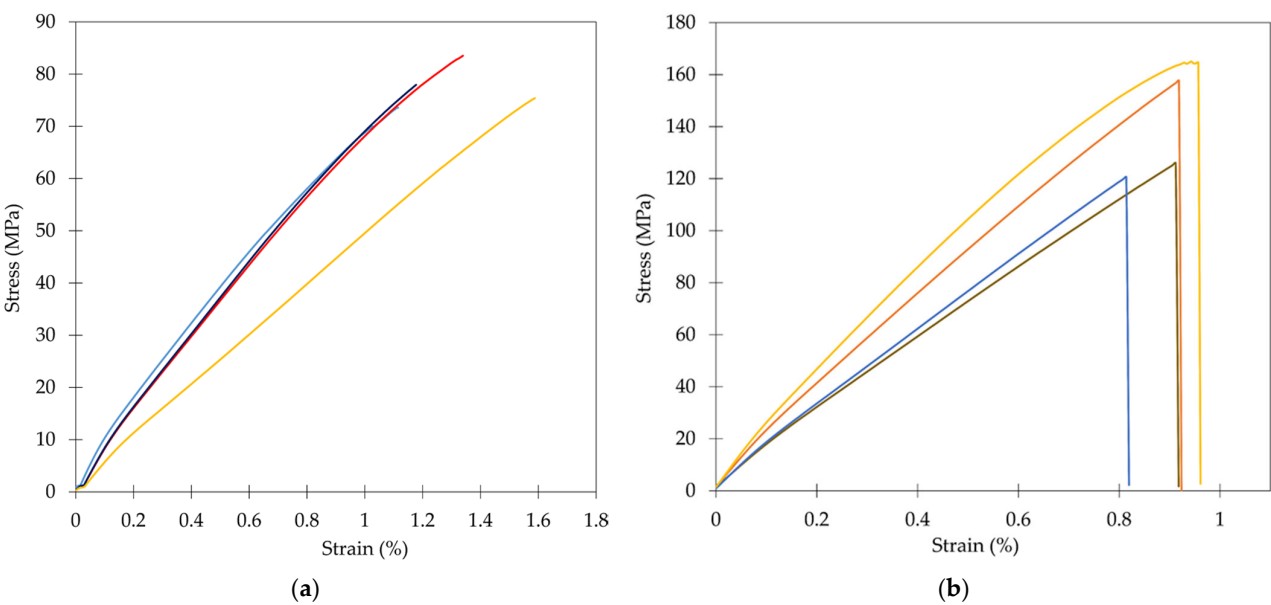

**Figure 16.** Tensile curves of manufactured composites: (**a**) CPP, (**b**) CQUD.

In Figure 17, the average properties of the CPP and CQUD composites are compared with those found in the literature for the same family: bio composites with reinforcement based on natural fiber made with epoxy resin. The characteristics and reference of the results shown in Figure 17 are detailed in Table 6. The first lines of Table 6 concern composites manufactured from woven fabrics, while the last lines concern composites from quasi-unidirectional fabrics. The tensile modulus of both composites—CQUD and CPP—was estimated in the strain range of 0.01% to 0.1%.

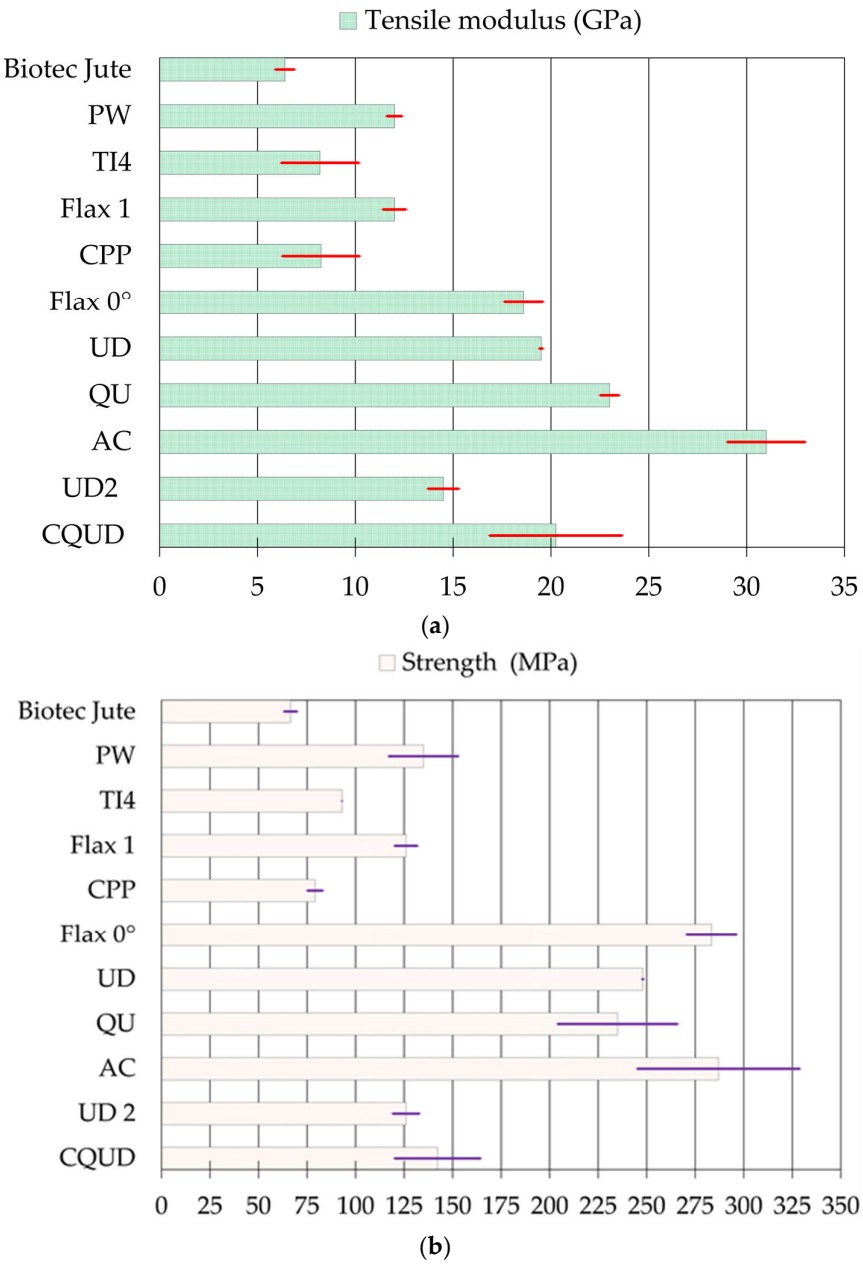

**Figure 17.** Comparison of tensile properties of CPP and CQUD composites: (**a**) tensile modulus; (**b**) strength at failure.

Compared to the properties obtained with woven fabric reinforcement, these results show that CQUD composites have very good stiffness and strength in their reinforcement direction (weft direction). These properties were better than those obtained in the warp direction of the CPP composite based on Plain_CL fabric, with differences of 44% and 60% for strength at break and stiffness, respectively. These results show that, at the composite

scale, the CQUD configuration provides the best properties in one direction. Similar results were obtained by Bensadoun et al. [6] for composites based on woven flax reinforcements, as well as by Corbin et al. [12] for hemp-based reinforcements.

**Table 6.** Characteristics of the study composites with biocomposites already studied in the literature and commercialized.

| Composite and References | Type of Reinforcement and Fibers | Areal Density (g/m$^2$) | Number of Plies | Fiber Content V$_f$ (%) | Manufacturing Process | Resin |
|---|---|---|---|---|---|---|
| Biotec Jute [46] | Fabric-plain weave jute | 500 | 4 | 40 | Infusion under vacuum (VARTM) | Epoxy SR 8100 of Sicomin |
| PW [6] | Fabric-plain weave flax | 285 | 4 | 40 | RTM | Epoxy Epikote 828 LVLE |
| TI4 [45] | Plain weave flax tows | 223.6 | 7 | 33 | Infusion under vacuum | Epoxy SR 8200/SD 820 |
| Flax 1 [7] | Fabric-plain weave flax | 308 ± 9 | 4 | 27.3 | Hot pressing | Green poxy 56 of Sicomin |
| CPP This study | Fabric-plain weave CL | 450 ± 25 | 2 | 29.62 | Hot pressing | Green poxy 56 of Sicomin |
| Flax 0° [46] | UD flax | 180 | 3 | 31 | Infusion under vacuum (VARTM) | Epoxy SR 8100 of Sicomin |
| UD [43] | Quasi-UD Twill 2 × 2flax | 217.8 ± 3.5 | 7 | 39 | Infusion under vacuum | Epoxy SR 8200/SD 820 |
| QU [6] | Quasi-UD plain weave flax | 300 | 4 | 40 | RTM | Epoxy Epikote 828 LVLE |
| AC [12] | Quasi-UD plain weave hemp | 649 ± 3 | 2 | 60 | Hot pressing | Green poxy 56 of Sicomin |
| UD2 [6] | Quasi-UD plain weaveflax | 200 | 4 | 40 | RTM | Epoxy Epikote 828 LVLE |
| CQUD This study | Quasi-UD plain weave CL | 385 ± 53 | 2 | 30.4 | Hot pressing | Green poxy 56 of Sicomin |

The CPP composite also has a higher strain at break than the CQUD composite due to the higher crimp of its woven reinforcement.

CPP composite manufactured from two layers of Plain_CL fabric, with a lower fiber content, has a higher tensile modulus and a higher stress at break than Biotec Jute (difference of 22.3% for tensile modulus and 16% for stress), as shown in Figure 17a. These results can be explained by the difference in tensile properties between jute and CL fiber [21,22]. There are few differences in tensile modulus and ultimate tensile stress values between the CPP sample (8.24 GPa stiffness and 79.14 MPa stress at failure) and the TI4 composite made from plain weave fabrics (warp/weft densities of 11 and 9, respectively) based on flax tows (linear density of 104 tex). However, the mechanical properties of the CPP composite are slightly lower than those of the PW and Flax 1 samples, which is related to their higher fiber content.

A comparison of the tensile properties of the CQUD composites with those of composites using quasi-unidirectional or unidirectional fabrics reported in the literature shows that the tensile modulus of the CQUD composites is, on average, similar. AC composites based on hemp roving [12] with a slightly higher fiber content have a 34% higher tensile modulus.

The lower linear density of hemp roving used in AC (~350 tex) compared to that of R1_CL roving (~830 Tex) used for CQUD, as shown in Table 2, may improve impregnation and, consequently, mechanical performances.

Concerning tensile strength, Figure 17b shows that the CQUD sample can be compared with composites based on woven fabrics, but its maximum tensile stress value remains lower than those of composites manufactured from unidirectional or quasi-unidirectional fabrics. The lower fiber content and the residual porosity obtained in this first manufacturing of composites based on CL_fibers can explain the results for tensile stress. The same observations can be made for the CPP composite samples in terms of their tensile strength, which remains smaller compared to other bio-based composites. The optimization of the process manufacturing, and the minimization of the crimp parameter at the scale of fabrics, would reduce the porosity rates and improve mechanical performances.

## 4. Conclusions

In this work, a multi-scale characterization of textile reinforcements for composite applications was performed. At the mesoscopic scale, rovings and yarns were manufactured from the CL ribbons obtained from wet retting of the bark of the plant. The characterization of textile and mechanical properties showed the potential of this tropical fiber to provide rovings and yarns suitable for weaving. The tenacities of twisted and untwisted CL yarns are higher than those of similar products based on flax or hemp fibers. Two types of fabrics were manufactured by weaving, at the lab scale, with significantly different weaving parameters, such as yarn density. Comparison of the properties of these reinforcements before impregnation showed that the quasi-unidirectional architecture allows crimp to be minimized and increases the orientation of the CL fibers. The deformability properties of both types of reinforcement architectures were also analyzed using a preforming device. Composite samples were made from these fabrics; the first tests showed the necessity of controlling the porosity level during the manufacturing steps, as well as the influence of the reinforcement characteristics on the mechanical properties. Despite the low fiber content of these first composite samples, the tensile properties compared to those of hemp- or flax-based composites are promising for the use of these tropical fibers in technical applications. In this experimental study, the properties identified at different scales also contribute to the valorization of these tropical fibers, which could also be used as reinforcements for pultruded composites [57–59].

**Author Contributions:** Investigation, R.L.N.; writing—original draft preparation, R.L.N.; validation and supervision, D.S., M.F., A.R.L., J.A.A. and L.M.; writing—review and editing, D.S., M.F. and A.R.L. All authors have read and agreed to the published version of the manuscript.

**Funding:** This research received no external funding.

**Data Availability Statement:** Not applicable.

**Conflicts of Interest:** The authors declare no conflict of interest.

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
