# Peer review of "Manufacture and Characterization of Cola Lépidota Reinforcements for Composite Applications"

_jcs, doi:10.3390/jcs7020065_

Round 1

Reviewer 1 Report

   This paper prepared two kinds of fabrics using a tropical fiber extracted from the bast of plant, and fabricated the composite using the fabrics. Mechanical and processing properties were characterized. Good performance was shown compared to those from the relevant reference. I suggest that it can been accepted after revision.

The advantages and drawbacks of various algorithms should be summarized. 

1. What do the unit Nmof linear density in Table 1 and tangent load in Figure 5 mean? 2. Why did not Fmax with hemispherical punch of Plain_CL in Table 3 be provided? 3. The resolution of the pictures in Figure 15 is low. 4. The title should be modified. Although the properties of roving, fabric and composite were measured, “multiscale characterization” in the title is not suitable. 5. “to the authors' knowledge, had not been done previously” and future work should be not given in the conclusion.

6. English grammar need to be polished.    

Author Response

Dear Editors and Reviewers,

Thank you for your letter and the Reviewers’ comments concerning our manuscript entitled “Manufacturing and characterization of Cola Lépidota reinforcements for composite applications. We have revised the manuscript carefully according to the Reviewer’s comments. The modification sentences are marked with yellow background in our revised manuscript and responses to Reviewers are (in red), in the file joined.

Best regards

For the authors : D. Soulat

Reviewer 2 Report

please, see the attachment

Author Response

Dear Editors and Reviewers,

Thank you for your letter and the Reviewers’ comments concerning our manuscript entitled “Manufacturing and characterization of Cola Lépidota reinforcements for composite applications. We have revised the manuscript carefully according to the Reviewer’s comments. The modification sentences are marked with yellow background in our revised manuscript and responses to Reviewers are  (in red) in the file joined

Best regards

For the authors, D. Soulat

Round 2

Reviewer 1 Report

The revision is good.

Reviewer 2 Report

All major comments were adequately addressed and the Authors have done an admirable job of improving the quality of the manuscript. Therefore, it can be accepted without any structural modification.